# Antifungal and Antibiofilm Activity of Cyclic Temporin L Peptide Analogues against Albicans and Non-Albicans *Candida* Species

**DOI:** 10.3390/pharmaceutics14020454

**Published:** 2022-02-21

**Authors:** Rosa Bellavita, Angela Maione, Francesco Merlino, Antonietta Siciliano, Principia Dardano, Luca De Stefano, Stefania Galdiero, Emilia Galdiero, Paolo Grieco, Annarita Falanga

**Affiliations:** 1Department of Pharmacy, School of Medicine, University of Naples ‘Federico II’, Via Domenico Montesano 49, 80131 Naples, Italy; rosa.bellavita@unina.it (R.B.); francesco.merlino@unina.it (F.M.); stefania.galdiero@unina.it (S.G.); 2Department of Biology, University of Naples ‘Federico II’, Via Cinthia, 80126 Naples, Italy; angela.maione@unina.it (A.M.); antonietta.siciliano@unina.it (A.S.); 3Institute of Applied Sciences and Intelligent Systems, Consiglio Nazionale delle Ricerche, Via Pietro Castellino 111, 80131 Naples, Italy; principia.dardano@cnr.it (P.D.); luca.destefano@cnr.it (L.D.S.); 4Department of Agricultural Science, University of Naples ‘Federico II’, Via Università 100, 80055 Portici, Italy

**Keywords:** biofilm, temporin L, helical peptides, *Candida* strains, toxicity, *Galleria mellonella*

## Abstract

Temporins are one of the largest families of antimicrobial peptides with both anti-inflammatory and antimicrobial activity. Herein, for a panel of cyclic temporin L isoform analogues, the antifungal and antibiofilm activities were determined against representative *Candida* strains, including *C. albicans*, *C. glabrata*, *C. auris*, *C. parapsilosis* and *C. tropicalis*. The outcomes indicated a significant anti-candida activity against planktonic and biofilm growth for four peptides (**3**, **7**, **15** and **16**). The absence of toxicity up to high concentrations and survival after infection were assessed in vivo by using *Galleria mellonella* larvae, and the correlation between conformation and cytotoxicity was investigated by fluorescence assays and circular dichroism (CD). By combining fluorescence spectroscopy, CD, dynamic light scattering, confocal and atomic force microscopy, the mode of action of four analogues was hypothesized. The results pinpointed that peptide **3** emerged as a non-toxic compound showing a potent antibiofilm activity and represents a promising compound for biomedical applications.

## 1. Introduction

Biofilms are aggregates of microorganisms or multicellular communities which embed themselves within a protective matrix of secreted extracellular polymeric substances, typically consisting of polysaccharides, proteins or peptides, lipids and DNA. Among all the microorganisms, fungal biofilms are highly associated with persistent nosocomial infections, most commonly typical in immunocompromised patients due to constant contact with medical devices including catheters, cardiac pacemakers and prosthetic heart valves [1]. Fungal biofilms are recognized as a complication of fungal infection treatments, which allows them to persist and trigger chronic infections, causing a dramatic rise in morbidity and mortality rates. Several clinically important species belonging to the genus *Candida* [2], including *C. albicans*, *C. auris, C. glabrata*, *C. parapsilosis* and *C. tropicalis* [3], are responsible for fungal biofilm-associated infections in the hospital setting [4]. *C. albicans* is the most common opportunistic pathogen, being endowed with a strong ability to switch from a yeast mode of growth to a hyphal morphology more adapted to the invasion of host tissues and able to form biofilms on abiotic and biotic surfaces, with increased recalcitrance and tolerance to existing antifungal agents [5,6,7]. Other non-*albicans Candida* species such as *C. auris*, *glabrata*, *tropicalis* and *parapsilosis* have been isolated from patients and are highly resistant to antifungals [8]. *C. tropicalis* is particularly relevant in urinary tract infections, while *C. parapsilosis* and *C. auris* are frequently found in the skin of healthy hosts, being the causative agent of catheter-related infections [9,10], *C. glabrata* is responsible for an estimated death rate of 40–60% after invasive candidiasis [11,12]. Although biofilms are formed by all the *Candida* species described above, they differ significantly from species to species in terms of morphology, characteristics of the extracellular matrix and ability to cause antifungal resistance. For example, *C. glabrata* biofilms are made of an intimately packed multilayer structure, while *C. tropicalis* biofilms are characterized by a network of yeast, pseudo-hyphae and hyphae with intense hyphal budding. Instead, *C. parapsilosis* biofilms are formed by clusters of yeast cells adhered to the surface with a minimal extracellular matrix [2]. These differences and variabilities highlight the complexity of the processes and the great challenge of finding an effective solution to tackle the threats of *Candida* biofilms. Due to the emergence of drug-resistant strains of *Candida*, the repertoire of antifungal agents is even more restricted. In fact, only three classes of drugs are available and effective to treat systemic fungal infections, namely polyenes (e.g., amphotericin B), triazoles (e.g., fluconazole), and echinocandins (e.g., caspofungin) [13]. Unfortunately, most of these antifungal agents are unable to eliminate the biofilm-associated infections, as the fungal cells within biofilm have shown a high level of drug resistance due to several mechanisms, such as the development of subpopulations of persister cells, the expression of genes encoding efflux pumps, and changes in the sterol composition of the cell membrane [14]. Polyenes display good activity in the treatment of biofilm-associated infections but require high doses that are considered toxic and unsafe [15]. Consequently, the search for alternative drugs that could disperse and eliminate biofilms, specially targeting the major virulence attributes of *Candida* species, is really unavoidable [16]. In this context, antimicrobial peptides (AMPs) with broad-spectrum activities represent excellent candidates for inhibiting fungal biofilm formation and eradicating preformed biofilms, thus being a promising strategy for novel therapeutic developments [17,18,19]. AMPs are produced from several synthetic and natural sources (such as vertebrates and insects) and demonstrate broad spectrum antimicrobial activity with high specificity and low toxicity. These peptides possess distinctive features such as usually being positively charged and assuming an amphiphilic structure in the membrane bilayer, and they function by employing sophisticated mechanisms of action correlated to a membrane interaction or intracellular mechanisms, such as gene regulation and generation of reactive oxygen species [20,21]. Temporins constitute one of the largest families of AMPs originally isolated from the mucus secreted from the granular skin glands of the red frog *Rana temporaria* [22,23]. Structurally, they are small-sized peptides (10–14 amino acids) featured by a net positive charge due to the presence of 2 or more basic residues and adopt an α-helical-like preferential conformation in a membrane environment. Among the numerous isoforms of this family, the membrane-active temporin L (TL, aa sequence: FVQWFSKFLGRIL) was extensively studied for its potent activity against Gram-positive and Gram-negative strains with minimal inhibitory concentrations (MICs) ranging from 2.5 to 20 μM. Interestingly, TL displayed a strong efficacy against yeast strains, including different *Candida* species such as *C. albicans* and *C. tropicalis* [24,25]. With the attempt to improve its therapeutic index through reducing its cytotoxicity toward mammalian cells, TL was subjected to several structure–activity relationship (SAR) studies. SAR studies, consisting of the application of different synthetic strategies, mostly focused on the design of analogues endowed with improved biological performance in terms of antimicrobial and cytotoxic activities, and their mechanism of action was extensively explored by using bacterial membrane mimetic systems [26,27,28,29]. Although certain studies focused on the investigation of a potential application of temporin-like peptides in the treatment of fungal infections [28], only a few TL analogues featuring significant antifungal and low cytotoxic activities were discovered [30]. Based on these findings and the key role of the α-helical content for biological activity, we envisaged the anticandidal activity for a panel of cyclic TL analogues, which we recently reported [31]. These peptide analogues were thus tested for their potential anti-candida activity against planktonic and biofilm growth from different *Candida* species. Additionally, the cytotoxicity and the antifungal potential use of the most promising peptides were assessed in vivo in *G. mellonella* larvae, and the correlation between the helical secondary structure and cytotoxicity was investigated by fluorescence assays and circular dichroism (CD) analyses. The eventual detection of mutagenesis by the most promising peptides in prokaryotic organisms was evaluated with a *Salmonella typhimurium* mutagenicity test. Moreover, our efforts were aimed at characterizing the mechanism of interaction through optical and atomic force microscopies, dynamic light scattering, fluorescence assays and CD all in liposomes mimicking fungal membranes.

## 2. Materials and Methods

### 2.1. Materials

All *N^α^*-Fmoc-protected conventional amino acids were acquired from GL Biochem Ltd. (Shanghai, China). Fmoc-Lys(Alloc), Fmoc-dLys(Alloc), Fmoc-Glu(OAll)-OH, Fmoc-dGlu(OAll)-OH, Fmoc-Pra-OH, Fmoc-dPra-OH, Fmoc-Lys(N_3_), Fmoc-dLys(N_3_), Rink amide resin (loading substitution of 0.72 mmol/g), *N,N*-diisopropylethylamine (DIEA), piperidine and trifluoroacetic acid (TFA) were purchased from Iris-Biotech GmbH (Marktredwitz, Germany). Coupling reagents such as morpholino-carbenium hexafluorophosphate (COMU) and ethyl cyano(hydroxyimino)acetate (Oxyma, Rotterdam, The Netherlands), Thioflavin T, Nile red and Triton X-100 were obtained from Sigma-Aldrich (St. Louis, MO, USA)/Merck (Kenilworth, NJ, USA). Moreover, peptide synthesis solvents *N,N*-dimethylformamide (DMF), dichloromethane (DCM), diethyl ether (Et_2_O), water and acetonitrile (MeCN) for HPLC were of a reagent grade, acquired from commercial sources (Sigma-Aldrich and VWR, Radnor, PA, USA) and used without further purification. The phospholipids of cholesterol (Chol), 1-Palmitoyl-2-oleoyl-*sn*-glycero-3-phosphocholine (DOPC), l-α-phosphatidylethanolamine (PE), l-phosphatidylglycerol (PG) and phosphatidylinositol (PI) were purchased from Avanti Polar Lipids, Inc. (Alabaster, AL, USA). Ergosterol was acquired from Iris-Biotech GMBH, while 8-aminonaphtalene-1,3,6-trisulfonic acid, disodium salt (ANTS) and p-xylene-bis-pyridinium bromide (DPX) were purchased from Molecular Probes.

### 2.2. Peptide Synthesis

All linear counterparts of helical peptides 1–17 were synthesized by ultrasonic-assisted solid-phase peptide synthesis (US−SPPS) combined with an Fmoc/tBu strategy [32] as reported elsewhere [31]. Briefly, each linear peptide was assembled on Rink amide resin by repeated cycles of Fmoc deprotections (20% piperidine in DMF, 0.5 + 1 min) and the coupling reactions (Fmoc-aa (2 equiv), COMU (2 equiv), Oxyma (2 equiv), DIPEA (4 equiv), 5 min in DMF). Upon the elongation of the linear precursor, depending on the type of cyclization, the synthesis proceeded according to procedures described elsewhere [31]. Finally, peptides 1–17 were purified and characterized by high-resolution mass (HRMS) spectrometry (see Appendix A) and RP-HPLC using linear gradients of MeCN (0.1% TFA) in water (0.1% TFA) from 10 to 90% over 30 min (see Appendix A).

### 2.3. Fungal Culture Conditions

*C. albicans* ATCC 90028, *C. auris* DSM 21092, *C. glabrata* DSM 11226, *C. tropicalis* DSM11951 and *C. parapsilosis* DSM 4874 grown on YPD Agar (1% *w*/*v* yeast extract, 2% *w*/*v* peptone, 2% *w*/*v* glucose, 1.5% Agar) were cultured in Tryptone soya broth supplemented with 0.1% glucose for 16–18 h at 37 °C, washed twice using sterile phosphate-buffered saline (PBS) and standardized to 106 cells/mL^−1^ as required for the experiments. RPMI1640 medium (Thermo Fisher Scientific, Waltham, MA, USA) buffered to a pH of 7.0 with 0.165 M MOPS was used for growing the biofilms of all *Candida* species.

### 2.4. Determination of Minimal Inhibitory Concentration (MIC) and Minimal Fungicidal Concentration (MFC)

The antifungal activity of peptides against all five *Candida* species was determined according to CLSI-M27-A3 [33], as reported previously [34]. Briefly, the compounds were tested at concentrations ranging from 6.5 to 50 μM, and the organism suspensions were adjusted to an inoculum of 10^6^ cell/mL at OD 590 nm in RPMI 1640, added to the 96-well plate and incubated at 37 °C for 24 h. The minimum inhibitory concentration (MIC) values were determined as the lowest concentration inhibiting fungal growth at 590 nm using a microplate reader (Synergy™ H4; BioTek Instruments, Inc., Winooski, VT, USA). The minimum fungicidal concentration (MFC) values were defined as the lowest concentration that showed no colony growth on the culture medium and were determined by subculturing 10 μL of the medium collected from the wells showing no microscopic growth on the YPD after 24 h. The MFC was the lowest concentration that yielded no colony growth on the agar. The MFC/MIC ratio was calculated to determine whether the substance had fungistatic (MFC/MIC ≥ 4) or fungicidal (MFC/MIC < 4) activity.

### 2.5. Fungal Biofilm Formation

To evaluate the ability to form *Candida* species biofilms, cell suspensions of each candida (10^6^ cell/mL) prepared in RPMI were placed in 96-well polystyrene microtiter plates (100 µL per well) and incubated for 24 h at 37 °C. After incubation, the medium was removed, and the biofilms were washed with 200 mL of phosphate-buffered saline (PBS) to remove non-adherent cells. The total biomass of the biofilms was quantified using the crystal violet (CV) staining method. The absorbance was measured spectrophotometrically at 570 nm [35,36].

### 2.6. Antibiofilm Activity

Biofilm inhibition and eradication assays were performed as previously described in [37]. Briefly, for the biofilm inhibition assay, each strain was incubated with sub-MIC concentrations of compounds ranging from 6.25 to 25.0 µM and incubated for 24 h at 37 °C. After the incubation period, non-adherent cells were detached by washing each sample three times in sterile PBS. Then, the samples were fixed, and the biofilms were stained with a 0.1% solution of crystal violet. To test the effect against the established biofilm, fresh growth medium containing peptides at the same concentrations was added in the preformed 24-h old biofilms and incubated for another 24 h. The total biomass was quantified by a crystal violet assay for the inhibition. The vital biomass was quantified by XTT for test eradication by using the tetrazolium 2,3-bis (2-methoxy-4-nitro-5 sulfophenyl)-5-[(phenylamine) carbonyl]-2H-hydroxide reduction assay (XTT) (Sigma-Aldrich, St. Luis, MO, USA) according to the manufacturer’s instructions. The absorbance was measured spectrophotometrically at 492 nm. The percentages of inhibition or eradication were calculated as follows: % biofilm reduction = Abs control − Abs sample/Abs control × 100.

### 2.7. Bright-Field Microscopy

Optical microscopy images were acquired by using a Leica AF6000LX-DM6M-Z microscope (Leica Microsystems, Mannheim, Germany) controlled by LAS-X (Leica Application Suite; rel. 3.0.13) software and equipped with a Leica Camera DFC7000T. Images were acquired by focusing on the Petri dish surface with a 50× objective in a bright field. A minimum of 10 images was acquired from 2 independent samples. The images had a resolution of 1920 × 1440 pixels.

### 2.8. Effect of Peptide Toxicity on G. mellonella

Twenty larvae with an average weight of 180–200 mg were used in each group in all assays with 2 control groups; one group was inoculated with PBS, while the other group received no injection as a control to evaluate general viability. A volume of 10 µL of each peptide at different concentrations (12.5, 25.0, and 50.0 µM) was inoculated in each group. All larvae were incubated at 37 °C and monitored daily for survival for 4 days in 3 independent experiments [34].

### 2.9. Kaplan–Meier Survival Analysis

The *G. mellonella* fungal infection model was performed as described previously [37]. We first determined the lethal concentrations of *C. albicans*, *C. auris*, *C. glabrata*, *C. tropicalis* and *C. parapsilosis* inocula by injecting concentrations of 10^4^, 10^5^ or 10^6^ cells/larvae. The larvae were kept in Petri dishes at 37 °C, and survival was monitored for 3 d. A system of scoring already proposed by Loh et al. [38] based on survival, mobility, degree of melanization, and the ability to produce a cocoon was used to evaluate the physical state of the larvae for 3 days [39]. To evaluate the effects of the five peptides on *Candida* infection and compare mortality, 10 μL of 10^6^ yeast cells in a medium containing 12.5 μM of each peptide’s concentrations was injected into the last left proleg of the larvae and incubated at 37 °C. The larvae were incubated at 37 °C and monitored daily for survival. The larvae were considered dead when they displayed no movement in response to being touched.

### 2.10. Mutagenicity with S. typhimurium (Fluctuation Ames Test)

The fluctuation Ames test was performed according to the manufacturer’s protocol provided for the Muta-ChromoPlate Bacterial Strain Kit (EBPI, Canada) and according to OECD Guideline 471 (OECD Guidelines for the Testing of Chemicals, Section 4, Test No. 471: Bacterial Reverse Mutation Test). *S. typhimurium* strain TA100 was used to assess the bacterial reverse mutation from amino acid (histidine) auxotrophy to prototrophy after exposure to mutagens. The overnight TA100 cultures were exposed to each peptide at a 12.5-µM concentration in the presence of bromocresol purple and then dispensed into 96-well microtiter plates. The plates were incubated at 37 °C for 5 days. After this period, the number of positive colonies was counted and statistically compared with the number of positive colonies in the blank wells to identify the spontaneous mutation rate. The negative control was distilled water, while the positive control was sodium azide for TA100 without S9. All experiments were conducted in duplicate. The results were expressed as the mutagenicity ratio (MR) and were obtained by dividing the number of reverse mutation colonies by the spontaneous mutation rate. When the MR was ≥2, the sample was judged to be mutagenic.

### 2.11. ANTS/DPX Leakage

ANTS/DPX-entrapped large unilamellar vesicles (LUVs) composed of PE/PC/PI/Ergosterol (5:4:1:2, *w*/*w*/*w*/*w*) and Chol/DOPC (70:30, ratio in moles) were prepared by the extrusion method as previously described in [28]. First, the lipid films were prepared by dissolving lipids in the proper ratio in chloroform to have a final concentration of 0.1 mM and then dried under a nitrogen gas stream. The probes of ANTS (12.5 mM) and DPX (45 mM) dissolved in water were added to the lipid films, and the mixture was freeze-dried overnight. The lipid films with encapsulated ANTS and DPX were hydrated with a PBS 1× buffer (pH = 7.2), vortexed for 1 h, freeze-thawed 6 times and extruded 10 times through 0.1-μm pore size polycarbonate filters to obtain the LUVs. The LUVs with entrapped ANTS and DPX were separated from the free ANTS and DPX by gel filtration chromatography on a Sephadex G-50 column (1.5 × 10 cm) [40]. The assay was based on the measurement of dequenching of ANTS by DPX when the LUVs were incubated with peptide concentrations of 5, 10, 15, 20, 30 and 50 μM. The increase in ANT fluorescence was recorded at an excitation fluorescence at 385 nm (slit width: 5 nm) and a fluorescence emission at 512 nm (slit width: 5 nm) for 10 min after the addition of the peptide. The destruction of all LUVs and the complete release of ANTs were obtained after treatment with 10% Triton-X. The percentage of liposome leakage was calculated as %leakage = (F_i_ − F_0_)/(F_t_ − F_0_), where F_0_ represents the fluorescence of the intact LUVs before the addition of the peptide and F_i_ and F_t_ are the intensities of the fluorescence achieved after peptide and Triton-X treatment, respectively.

### 2.12. Zeta Potential Measured by Dynamic Light Scattering

The zeta potential of the fungal cells was recorded in a suspension (0.5 × 10^6^ cells/mL) using the electrophoretic light scattering technique on a Zetasizer Nano ZS analyzer [41]. Before the peptide treatment, the cells were suspended using a 0.05% trypsin solution in 0.53 mM EDTA. The concentration of the suspended cells in a buffered phosphate saline (PBS, 1.7 mM KH_2_PO_4_, 5.2 mM Na_2_HPO_4_ and 150 mM NaCl) was determined on a hemocytometer. The measurements were performed by diluting the cells with PBS 0.001× and treating them with the peptide at a concentration of 25 μM.

### 2.13. Critical Aggregation Concentration Calculated Using Nile Red as a Probe

The peptide aggregation in the aqueous solution was monitored using Nile red as a dye probe. Nile red has poor solubility in water, and it has a large tendency to move in a hydrophobic environment as aggregates, showing a blue shift [42]. First, the stock solutions of the peptides were prepared by dissolving them in ethanol, and then different aliquots were taken to prepare the peptide solutions at different concentrations (1, 5, 10, 15, 20, 30, 50, 100 and 200 μM). The ethanol was evaporated under a nitrogen gas stream and then dissolved with sterile water (1.5 mL), sonicated for 15 min and freeze-dried. Then, each peptide solution was hydrated with Nile red solution (C = 500 nM) in the dark place for 1 h before measurement. The emission spectra of Nile red were measured between 570 and 700 nm at a slit width of 5 nm using an excitation wavelength of 550 nm and a 10-nm slit width. The data were repeated in triplicate and analyzed by plotting the maximum emission fluorescence corresponding wavelength (y) as a function of the peptide concentration (***x***) and fitting with the sigmoidal Boltzmann equation:$$y=\frac{A1-A2}{1+e^{\left(x-\frac{x0}{\Delta x}\right)}}+A2$$

In the equation, *A*1 and *A*2 indicate the upper and lower limits of the sigmoid, respectively, *x*_0_ is the inflection point of the sigmoid and Δ*x* is the parameter which represents the steepness of the function. The sigmoidal plot allows for calculating the CAC value at *x*_0_.

### 2.14. Peptide Aggregation Monitored by a Thioflavin T (ThT) Fluorescence Assay

The aggregation in the liposomes mimicking fungal membranes (5:4:1:2, *w*/*w*/*w*/*w*) was evaluated using Thioflavin T (ThT) as a dye probe [43]. The lipid film was prepared by dissolving lipids in chloroform to have a final lipid concentration of 0.1 mM. Then, the solvent was removed under a nitrogen gas stream, and the film was freeze-dried overnight. Lipid films were hydrated with 100 mM NaCl, 10 mM Tris-HCl and a 25-mM ThT buffer at a pH of 7.4 and then treated as described above to obtain LUVs. The LUVs were titrated with a peptide concentration of 5, 10, 15, 20, 30 and 50 μM, and ThT fluorescence was recorded before and after the addition of the peptide using the Varian Cary Eclipse fluorescence spectrometer at 25 °C. The samples were excited at 450 nm (slit width: 10 nm), and the fluorescence emission was recorded at 482 nm (slit width: 5 nm). Aggregation was quantified according to the equation %A = (F_f_ − F_0_)/(F_max_ − F_0_) × 100, where F_f_ indicates the value of fluorescence after the peptide addition, F_0_ is the initial fluorescence before the addition of the peptide and F_max_ is the fluorescence maximum obtained immediately after the peptide addition.

### 2.15. Secondary Structural Analysis

The circular dichroism spectra of the selected peptides (3, 7, 15 and 16) were recorded in small unilamellar vesicles (SUVs) composed of PE/PC/PI/Ergosterol (5:4:1:2, *w*/*w*/*w*/*w*, final lipid concentration = 0.01 mM) and DOPC/Chol (70:30, ratio in moles, final lipid concentration = 0.1 mM), between 190 and 260 nm on a JASCO J-710 CD spectropolarimeter at 25 °C using quartz cells with a 0.1-cm path length. The SUVs were prepared as reported in the following protocol [28]. Once the lipid films were prepared as described above, they were hydrated with a phosphate buffer (5 mM, pH 7.4) (DOPC:Chol) or water (PE/PC/PI/Ergosterol) for 1 h and sonicated for 40 min. The peptides were dissolved in water to form a sample with a concentration of 80 μM in SUVs made of DOPC/Chol, while a peptide concentration of 10 μM was used in SUVs made of PE/PC/PI/Ergosterol. We accumulated the data three times, and all spectral data were subtracted from the background. Each spectrum converted the signal to the mean molar ellipticity, and we calculated the ratio of the ellipticities at 222 and 208 nm to discriminate between the monomeric and oligomeric states of the helices. The helix content was calculated as follows: [θ]_222_/(−40,000[1 − 2.5/*n*)], where [θ]_222_ is the ellipticity measurement at 222 nm and *n* is the number of amino acid residues [44].

### 2.16. Statistical Analysis

All statistical analyses were performed using GraphPad Prism software (version 8.02 for Windows, GraphPad Software, La Jolla, CA, USA). A *t*-test was used to assess the statistical differences from the control group (*p* < 0.05). Statistical significance between different groups was performed by two-way analysis of variance (ANOVA) followed by a post hoc Tukey’s test for multiple comparisons (*p* < 0.05). The Kaplan–Meier method and log-rank test were used in the *G. mellonella* model to plot survival curves.

### 2.17. Atomic Force Microscopy Imaging

XE-100 Atomic Force Microscopy (AFM) (by Park Systems, Suwon, Korea) was used for the imaging of the most promising peptide (3) with and without liposome interaction. Surface imaging was obtained in non-contact mode using SSS-NCHR 10 M (by Park Systems) Si/Al-coated cantilever with a sharp tip (typical tip radius: 2 nm, <5 nm max), a resonance frequency between 200 and 400 kHz and a nominal force constant of 42 N/m. The images had a 2064 × 2064 pixel resolution, and the used scan frequency was 0.5 Hz per line. The AFM images were processed and analyzed by the home software (XEI by Park Systems). The images were flattened to remove the background slope and adjusted in terms of contrast and brightness. For the sample preparation for AFM, the AFM imaging was performed on a substrate of a Muscovite mica surface (of about 1 cm^2^ in area) that owned a root mean square (rms) surface roughness of less than 0.5 nm on a 1000 × 1000 nm^2^ area. The 3-µL aliquots of the sample solutions were casted on mica immediately after cleavage with adhesive tape to establish their cleanliness and charge activity. In fact, after the cleavage, the K^+^ ions bonding the mica layers were highly mobile, resulting in a positive overcharging of the mica surface, which enabled the deposition of molecules that held a negative charge [45]. After 2 min, every sample was gently washed with deionized water to remove the sample excesses and then dried at room temperature under a ventilated fume hood.

## 3. Results

### 3.1. Peptide Synthesis

Peptides **1**–**17** (Table 1) were synthesized by applying the Fmoc-based ultrasonication-assisted solid-phase peptide strategy (US-SPPS) [32]. Upon the elongation of the linear precursors, reactions to yield lactam (**1**–**5**, **11**, **12** and **15**)-, 1,4-triazolic (**6**–**10**, **13** and **14**)-, disulfide (**16**)-, and olefin (**17**)-bridged peptides were performed according to procedures previously reported [31].

### 3.2. Peptide Screening by MIC for Candida Species

A comprehensive screening of the antifungal activity of peptides was performed using *C. albicans* ATCC 90028 and *C. parapsilosis* DSM5784 in vitro. All peptides showed activity against fungal cells with a moderate MIC greater than 50 μM, with the exception of peptide **3**, bearing the lactam bridge between positions 6 and 10, peptide **7** with a 1,4-triazolic bridge between positions 8 and 12, peptide **15** featured by a reverse orientation of the lactam bridge compared with peptide 3, and peptide **16** featuring a disulfide bridge between positions 6 and 10 (Table 1). In particular, peptide **15** presented inhibitory effects toward *C. parapsilosis* at a concentration of 50 μM and over 50 μM for *C. albicans*, while peptides **3**, **7** and **16** displayed an excellent MIC for both strains.

Therefore, peptides **3**, **7**, **15** and **16** were considered the most promising peptides and were used in subsequent experiments on *C. albicans*, *C. parapsilosis*, *C. auris*, *C. glabrata* and *C. tropicalis*. The antifungal activity of these selected peptides was significant for all the *Candida* strains explored (Table 2), with the sole exception of peptide **15**, which showed an MIC >50 μM against *C. albicans* and was ineffective up to a minimum fungicidal concentration (MFC) >50 μM. Peptide **16** was the most active on *C. glabrata* and *C. tropicalis*, with an MIC of 25 μM. For all *Candida* strains, the MFC values were similar to the MIC values, indicating their significant killing capacity (Table 2).

### 3.3. Activity against Candida Species Biofilm for a Selection of Peptides **3**, **7**, **15** and **16**

The biomass of each candida biofilms (*n* = 3, mean ± SD) was quantified by crystal violet staining and expressed as OD570 (Figure 1).

All fungi were able to form in vitro biofilms based on the classification of adherence capabilities reported by Stepanovic et al. [35]. In particular, the adherence capabilities were classified as negative for OD ≤ ODc, weak for ODc ≤ OD ≤ 2 ODc, moderate for 2 ODc < OD ≤ 4 ODc and strong in biofilm production for 4 ODc < OD. As a matter of fact, *C. auris* and *C. tropicalis* were defined as strong, *C. albicans* and *C. parapsilosis* were moderate, and only *C. glabrata* was considered a weak biofilm producer. Since peptides **3**, **7**, **15** and **16** displayed a significant antifungal activity against all five *Candida* species, their activity against biofilms was further investigated. Usually, AMPs can prevent or eradicate microbial biofilms independently of their antimicrobial activity against free planktonic microorganisms. Thus, we used sub-MIC concentrations to evaluate the efficacy of our most promising peptides in inhibition and eradication assays. Figure 2 shows a drastic reduction of the biofilm development for all *Candida* species at all concentrations tested. At the highest concentration of 25 μM, the reduction was from 50% to 90% for all peptides, while at the lowest concentration of 6.25 μM, peptide **3** reached 50% in *C. parapsilosis* and in *C. auris*, and peptide **16** reached about 50% in *C. tropicalis*. The eradication effects of the four peptides on the preformed biofilms were significant principally for peptide **3**, with about 90% eradication toward all *Candida* species at the highest concentration tested (Figure 3). At the concentration of 6.25 μM, peptide **3** still presented a 50% eradication capability for *C. albicans* and *C. parapsilosis*. Peptide **7** showed a good eradication efficacy at the lowest concentration between 30 and 50% for *C. albicans* and *C. parapsilosis.* In addition, the eradication effect of peptide **3** on the preformed *C. albicans* biofilms was also evaluated by optical microscopy in a bright field (Figure 3f). The treatment of the *C. albicans* biofilms with peptide **3** at a concentration of 12.5 μM was proven to be effective in the eradication of biofilm and mainly in inhibiting hyphal growth, which was instead clearly visible in the control culture.

### 3.4. Peptide Toxicity against G. mellonella Larvae

A *G. mellonella* larva is an excellent model organism for in vivo toxicology and pathogenicity testing, replacing the use of small mammals for initial screening thanks to its significant ethical, logistical and economic advantages [46]. Thus, the toxicity of the most promising peptides (**3**, **7**, **15** and **16**) was assessed in *G. mellonella* larvae (Figure 4). No larval killing was observed in the control larvae injected with an equivalent volume of PBS. Peptide **3** showed no significant toxic effect at any of the concentrations tested, while the other peptides already exhibited a greater toxicity at the concentration of 50 μM at 24 h from injection.

### 3.5. Peptide Effect on Infected G. mellonella Larvae

The in vivo effectiveness of peptides **3**, **7**, **15** and **16** was evaluated on infected *G. mellonella* larvae by different *Candida* species (Figure 5). In our study, we induced *C. albicans*, *C. auris*, *C. glabrata*, *C. tropicalis* and *C. parapsilosis* infection in the larvae, and the characteristics of each disease were considered as proposed by Loh et al. [38].

### 3.6. Peptides Did Not Induce Mutagenesis in Ames Test

To evaluate the mutagenic potential of peptides **3**, **7**, **15** and **16**, we used the mutation assay in *S. typhimurium* (Ames test), which is a bacterial short-term test established by regulatory agencies worldwide and used to identify carcinogens that can provoke damage to DNA, inducing gene mutations. Within the current accepted in vitro genotoxicity tests, the Ames test represents the first step of the genotoxicity assessment, as it is useful for identifying substances that can elicit mutations in the early stages of product development. In this test, the *Salmonella* strains characterized by genetic mutations in the histidine operon, which make the bacteria unable to synthesize the histidine amino acid, are employed [47]. In our studies, the tested peptides were not potentially mutagenic at a concentration of 12.5 µM, as evidenced by the MR values reported in Figure 6. Specifically, the mutagenic ratio showed values <2, with a number of *S. typhimurium* revertants grown in the presence of the tested samples statistically lower than the number of revertants in the control group, except for peptide **16**.

### 3.7. Correlation between Structural Features and Eukaryotic Toxicity

To better understand the structural features contributing to toxicity for the eukaryotic cells, we used liposomes made of 1-Palmitoyl-2-oleoyl-*sn*-glycero-3-phosphocholine (DOPC) and cholesterol (Chol) (DOPC:Chol, 70:30, ratio in moles) that mimicked eukaryotic membranes. The structural changes of peptides **3**, **7**, **15** and **16** were investigated in DOPC:Chol liposomes using circular dichroism (Figure 7a), and the percentage of the helix was calculated according to the formula reported previously [44].

The CD spectra clearly showed that all peptides adopted an α-helix conformation revealed by the two minima close to 208 and 222 nm, except for the disulfide-bridged peptide **16** showing a random coil conformation. Peptide **15**, endowed with the lactam bridge with a reverse orientation compared with peptide **3,** displayed both the highest helical percentage (~48%) and the strongest propensity to form helical aggregates in the eukaryotic membranes, as evidenced by the ratio of the ellipticities at 222 and 208 nm greater than 1 [48]. The 1,4-triazolic-bridged peptide **7** also adopted a helix structure with a percentage of ~23%, followed by the lactam-bridged peptide **3** with a helical content of ~13%.

Furthermore, we performed a leakage assay by measuring the release of entrapped ANTS fluorophore to characterize the mechanism of membrane interaction. After the preparation of the LUVs, the peptides were added at different concentrations, and the change in ANTS fluorescence was recorded because of pore formation. The most cytotoxic lactam-bridged peptide (**15**) exhibited the strongest ability to cause leakage, triggering the complete disruption of liposomes at the highest concentration of 50 μM, as shown in Figure 7 (panel B). Instead, peptides **3**, **7** and **16** exhibited the same trend to generate eukaryotic membrane leakage in the concentration range used, just inducing ~28% liposome disruption at the highest concentration of 50 μM. The ability of the peptides to induce pore formation in the eukaryotic membranes was clearly correlated to the helical content and mode of action. Our data show that the leakage of the membrane was one possible mode of action that contributed to the cytotoxicity of these stapled peptides.

### 3.8. Non-Permeabilizing Action of Stapled Peptides in Fungal Membranes

To investigate the mode of action of the selected peptides (**3**, **7**, **15** and **16**) against *C. albicans*, we assessed their leakage capability by measuring the release of entrapped ANTS fluorophore from the liposomes made of l-α-phosphatidylethanolamine (PE), l-phosphatidylglycerol (PG), phosphatidylinositol (PI) and ergosterol (PE/PC/PI/ergosterol, 5:4:1:2, *w*/*w*/*w*/*w*), which mimicked the fungal cell membrane. As shown in Figure 8, peptide **3** caused ~30% fungal liposome leakage at a concentration of 20 μM, while a lower ANTS release was observed after the addition of peptides **7**, **15** and **16** at the same concentration.

Unlike the conditions used in the leakage experiment performed in eukaryotic membrane-like lipids, in this lipid condition, the measurements were not recorded at the highest peptide concentrations because macroscopic peptide aggregates were observed. Interestingly, these results suggested that their mode of action may involve other molecular mechanisms underlying their antifungal activity.

### 3.9. Peptide Interaction with Lipid Fungal Membranes

To evaluate the changes taking place on the fungal membrane after the treatment with peptides **3**, **7**, **15** and **16**, we calculated the zeta potential by dynamic light scattering in the presence of *C. albicans* cells (10^6^ cells mL^−1^). The calculated peptide charge was +3 for all the peptides used. Before the peptide treatment, the zeta potential value of the *C. albicans* cells was negative (−6.2 ± 0.5), and it became highly positive after incubation with peptides **3** (1.6 ± 0.3 mV) and **15** (13.5 ± 3.4 mV) at a concentration of 25 μM, indicating the peptide presence on the cell surface. In contrast, no significant alteration in the zeta potential of *C. albicans* exposed to peptides **7** (−4.5 ± 1.6 mV) and **16** (−6.5 ± 0.1) was observed at the same peptide concentration. Table 3 reports the theoretical zeta potential obtained by adding the charge of the peptides to the zeta potential of the cells and the experimental one obtained after treatment with the peptides. Clearly, the positive zeta potential was an indication of aggregation or the presence of several peptide molecules on the surface of the cells.

### 3.10. Peptide Aggregation in Different Environments

Since the peptide aggregation on the cell surface could promote changes in the zeta potential of fungal cells and may represent one possible mode of action that contributes to peptide activity, we examined the aggregation behavior of peptides **3**, **7**, **15** and **16** in an aqueous solution using Nile red as a dye probe (Figure 9a) and in LUVs mimicking fungal membranes (Figure 9b,c). First, the aggregation phenomenon was monitored in an aqueous solution, and the critical aggregation concentration (CAC) for each peptide was calculated, reporting the wavelength of Nile red maximum fluorescence emission as a function of the concentration. Peptides **3** and **15** showed a similar ability to aggregate in water with a CAC of 17 μM, while peptides **7** and **16** showed a low ability to aggregate in an aqueous solution with a higher CAC value of 40 μM (Figure 9b). In contrast, all peptides showed a high tendency to aggregate in fungal membrane-like lipid compositions (LUVs made of PE/PC/PI/ergosterol, 5:4:1:2, *w*/*w*/*w*/*w*). In particular, LUVs hydrated with buffer containing Thioflavin T (ThT) as a dye probe were titrated with peptides at different concentrations, and a large enhancement of ThT fluorescence was recorded at 20 μM, revealing a complete aggregation of each peptide except for peptide **7**, which was completely aggregated at 30 μM (Figure 9, panel c)**.** In addition, the peptide aggregation was confirmed by the CD analysis after the treatment of liposomes (SUVs, PE/PC/PI/ergosterol, 5:4:1:2, *w*/*w*/*w*/*w*) with peptides at concentrations of 10 μM and 50 μM (panels d and e, respectively). As highlighted in Figure 9 (panel d), all peptides showed a significant tendency to form α-helical molecular aggregates already at a concentration of 10 μM, indicated both by the two minima close to 208 and 222 nm and the ratio *θ*_222_/*θ*_208_ being greater than 1.0 [48], except for disulfide-bridged peptide **16**, which adopted a random coil conformation. In fact, peptide **16** tended to form α-helical molecular aggregates at a higher concentration of 50 μM as shown by panel E in Figure 9. Overall, the CD data clearly indicate that the peptides tended to aggregate because of the interaction with the membrane bilayer.

### 3.11. Peptide Aggregation Imaged by AFM

The aggregation in the presence of liposomes was further investigated by atomic force microscopy (AFM). Figure 10 shows the images obtained for peptide **3** at 50 μM before (Figure 10a) and after the interaction with liposomes (Figure 10b). It is evident that there was a different kind of aggregation when probing the peptide in the absence and presence of liposomes. In the absence of liposomes (Figure 10a), we observed some clusters up to 300 nm wide and 20 nm high deposited on the mica surface. The random shape and heights support the presence of a non-molecular precipitate on the mica surface. Interestingly, Figure 10b shows that in the presence of liposomes, peptide **3** arranged on the mica substrate formed a kind of thin layer (4 nm), likely with a strong interaction with the positively charged substrate. The data clearly indicate that in the presence of liposomes, peptide **3** was forming molecular aggregates, which could not be visualized by AFM but further supported the data obtained by CD spectroscopy.

Note that peptide **16** AFM imaging revealed no significant changes in the arrangement states on the mica substrate before (Figure 11a) and after (Figure 11b) interaction with the liposomes. This result confirms that no significant alteration in the charge of peptide **16** (see zeta potential results) occurred before or after liposome exposition. In fact, in both cases, the strong interaction between negatively charged peptide **16** and the positively charged substrate resulted in an arrangement on the mica substrate in the form of a thin layer (2 ÷ 3 nm).

## 4. Discussion

Biofilms are aggregates of microorganisms in which cells are embedded in a self-produced matrix of extracellular polymeric substances (EPSs) that adhere to each other or a surface and are frequently more resistant to antibiotics than planktonic bacteria or fungi [1]. Antibiotics routinely used for the treatment of planktonic infections are also applied against biofilms, often leading to antibiofilm therapy failure [13]. Moreover, biofilm cells can become recalcitrant, which hampers their eradication and treatment effectiveness.

AMPs represent an interesting class of molecules with potent activity against biofilms both in vitro and in vivo as well. Apart from preventing biofilm formation, they are also able to eradicate preformed biofilms [17,49]. It is also worth noting that some AMPs are not active against bacteria or fungi in their planktonic mode of growth but are significantly active against biofilms. In this study, a panel of cyclic temporins recently developed by us [31] was initially screened for their antifungal activity against two planktonic *Candida* species: *C. albicans* and *C. parapsilosis* (Table 2). The cyclic peptides **1**–**17** were featured by different chemical bridges. In particular, lactam- and 1,4-triazolic bridges were used for their known ability to stabilize helical structures [31]. The entire library of peptides resulted in being inactive with MIC > 50 μM for both species, except for peptides **3** and **15** both carrying the lactam bridge in the same *i,i* + 4 positions but in different orientations, peptide **7** bearing the 1,4-triazolic bridge in positions 8 and 12, lactam-bridged peptide **15** and disulfide-bridged peptide **16** showing a moderate MIC around 50 μM. We thus selected these four peptides as the most promising compounds and evaluated their antifungal activity against *C. albicans*, *C. parapsilosis*, *C. auris*, *C. glabrata* and *C. tropicalis* species (Table 3). All peptides displayed significant activity toward all *Candida* species. Interestingly, peptide **16** bearing the disulfide bridge ended up being the most active on *C. glabrata* and *C. tropicalis* with an MIC of 25 μM. After this screening of antifungal activity, the ability of peptides **3**, **7**, **15** and **16** to inhibit biofilm formation at sub-MIC concentrations was explored toward all *Candida* species. As shown in Figure 2, all peptides were able to inhibit the biofilm of all *Candida* spp. at a concentration of 25 μM. In particular, the lactam-bridged peptide **3** kept potent activity even at a lower concentration of 6.25 μM, promoting 50% inhibition on the *C. parapsilosis* and *C. auris* biofilms. While the prevention of biofilm formation is desirable and is a major topic of research, the removal of established biofilms remains a challenge. We exposed established biofilms to our analogues, and they were able to eradicate the preformed biofilms. Interestingly, the lactam-bridged peptide **3** also showed the highest eradication effect (90%) on preformed biofilms of all *Candida* species at the highest concentration tested, and it still presented a 50% eradication capability on *C. albicans* and *C. parapsilosis* at 6.25 μM (Figure 3). Furthermore, the eradication effect of peptide **3** on preformed *C. albicans* biofilms was also evaluated by optical microscopy in a bright field, confirming that the peptide was effective in the eradication of biofilm and in interfering with hyphal growth which, on the contrary, was clearly visible in the control culture. Our results support the view that the peptides were able to damage the structure of the biofilm and interfere with the fungal cells. Indeed, biofilm formation and life is a dynamic, multi-step process that begins with surface detection, followed by attachment to the surface. We thus hypothesized that the ability of the analogues to inhibit or eradicate biofilms could be due to the effect of the peptides on these surface interactions. Previous studies on cytotoxicity performed against human keratinocytes [31] showed that lactam-bridged peptide **3** is less cytotoxic even at a high concentration of 50 μM, while peptide **15** bearing the lactam bridge in a different orientation compared with peptide **3** was the most cytotoxic in vitro. In this study, we confirmed these results in vivo using *G. mellonella* larvae as an excellent in vivo model [38]. These experiments confirmed our previous results, with peptide **3** showing no significant toxic effect at all the concentrations tested, while the other peptides, especially peptide **15**, already exhibited a greater toxicity at the concentration of 50 μM 24 h after injection (Figure 4). Nonetheless, none of the peptides showed any mutagenic potential as evaluated with the *S. typhimurium* mutagenicity test (Ames test).

Moreover, peptides **3**, **7**, **15** and **16** showed an in vivo effectiveness on the infected *G. mellonella* larvae via different *Candida* species. In particular, the treatment with peptide **3** at 12.5 μM increased the survival rate of the larvae infected by all strains after 48 h.

The remarkable biological results prompted us to analyze through biophysical studies the mechanism of action underlying the cytotoxicity and antifungal activities of the four selected peptides. Although the AMP mechanisms of action are still not well known, they are likely to involve membrane disruption through a detergent-like membrane destabilization (similar to a carpet model) or pore formation (barrel-stave and toroidal models) [17]. We thus performed a deep biophysical analysis of the mechanism of activity or toxicity of our four analogues through the determination of their interaction with liposomes mimicking the membranes of fungi (activity) and eukaryotic cells (toxicity). In particular, cell membranes are mainly composed of lipids organized into two leaflets which separate the interior of the cells or cellular compartments from their outer environments, and the precise membrane lipid composition varies greatly in different organisms, tissues, cell types, compartments or organelles, and the lipid composition differences can be exploited for the selective targeting of peptide activity on specific membranes. Moreover, the activity can be very sensitive to changes in the peptide sequence, which can have a profound effect on the activity and on the mechanism of action.

To understand if the interaction between the cyclic temporin peptides and the fungi surface leads to membrane permeation, which is the AMPs’ most widely accepted mode of fungicidal action, we performed membrane leakage assays using liposomes mimicking the fungal membrane (PE/PC/PI/ergosterol, 5:4:1:2, *w*/*w*/*w*/*w*). We observed a significant leaking activity reaching approximatively 30% for lactam-bridged peptide **3** (Figure 8), which was still not extremely high and likely suggested a distribution of the peptide on the fungal surface and the local denaturation and permeabilization of the membrane. Interestingly, peptide **15**, bearing the lactam bridge in the reverse orientation with respect to peptide **3**, showed a lower leakage activity against fungal membranes (less than 20%), whereas it showed the greatest leakage activity (100%) against liposomes mimicking eukaryotic membranes at the highest concentration of 50 μM (DOPC:Chol, 70:30, *w*/*w*). Moreover, these permeabilization data obtained on liposomes mimicking fungal and eukaryotic membranes clearly agreed with the cytotoxicity results obtained for peptides **3** and **15** in both the in vitro and in vivo MTT experiment models of *G. mellonella* larvae.

Given that our results for liposomes mimicking a fungal membrane showed a significant but still weak ability for peptides **3**, **7**, **15** and **16** to induce leakage, we explored other features that could influence the fungicidal activity to hypothesize their mode of action. One is certainly the ability to undergo a conformational transition to α-helical structures when moving from the aqueous to the membrane environment, but the possibility to produce amphipathic structures, the aggregation phenomena and the presence of aromatic residues were also considered [28]. A higher helicity in the aqueous solution is often correlated to lower activity, while a greater ability to produce a helical structure in membranes is associated with greater activities. Furthermore, it is widely accepted that aggregation phenomena in an aqueous solution can cause a loss of antimicrobial or antifungal activity, leading to an increase in MIC, while aggregation in microbial membranes induce greater activity. We calculated the CAC for each peptide in the aqueous solution using Nile red as a fluorescent probe. Peptides **3** and **15** bearing the lactam bridge showed a similar ability to aggregate in water with a CAC of 17 μM, while 1,4-triazolic-bridged peptide **7** and disulfide-bridged peptide **16** showed a lower ability to aggregate in an aqueous solution with a higher CAC value of 40 μM. The aggregation phenomenon in the solution of the most active peptide (**3**) was investigated by AFM, showing the formation of some clusters up to 300 nm wide and 20 nm high that support the presence of a non-molecular precipitate on the mica surface. In contrast, all peptides were completely aggregated in the fungal membranes at 20 μM, revealed by a high enhancement of the ThT fluorescence. In addition, we decided to monitor the change of the secondary structures of peptides when moving from an aqueous environment to a membrane mimetic environment by CD analysis. While the CD spectra in water reflect a random structure for all peptides [31], on the contrary, when we added peptides below the CAC at 10 μM, we observed a change in the secondary structure of the peptides, leading to helical molecular aggregates characterized by a ratio *θ*_222_/*θ*_208_ greater than 1, except for disulfide-bridged peptide **16**, which assumed a random coil structure. In fact, unlike peptides **3**, **7** and **15**, peptide **16** showed a tendency to form helical aggregates at a concentration above the CAC of 50 μM. These data clearly indicate that the peptides tended to aggregate because of the interaction with the membrane bilayer. Noteworthy is the analysis of the CD spectra in liposomes mimicking eukaryotic membranes, which showed much a higher tendency of lactam-bridged peptide **15** to produce helical molecular aggregates, which further confirmed the different behavior of peptide **15** compared with peptide **3** toward eukaryotic membranes. Furthermore, the CD data clearly show that the most cytotoxic lactam-bridged peptide **15** had both the highest helical content (~48%) and the strongest propensity to form helical aggregates in eukaryotic membranes, while conversely, the less cytotoxic peptide **3** displayed ~13% of the helical content in the eukaryotic membranes. The structural results highlight the strong correlation between the leakage ability and the helical content. The peptide aggregation on the fungal cell surface was further evaluated by measuring the changes in the zeta potential of *C. albicans* cells by dynamic light scattering. As reported in Table 2, before peptide addition, the fungal cells possessed a net negative charge (−6.2 ± 0.5 mV), which changed to positive values after the addition of peptides **3** (1.6 ± 0.3 mV) and **15** (13.5 ± 3.4 mV). The results indicate that these peptides were certainly aggregating and covering the surface of the fungi.

Overall, by these intensive biophysical studies on peptide aggregation and a previous conformational analysis of lactam-bridged peptide **3** in a membrane mimetic environment performed by NMR spectroscopy [31], we hypothesized the mechanism of action of the most active peptide. The NMR study carried out in sodium dodecyl sulfate (SDS)/dodecylphosphocholine (DPC) (9:1, *v:v*) showed that peptide **3** assumed an α-helical structure from residue 5 to residue 13 and adopted an amphipathic structure with (1) a positive domain at the *C*-terminus, which was likely involved in the initial approaching of the microbial membrane, and (2) a hydrophobic domain with an aromatic residue at the *N*-terminus (Phe), which was likely involved in the penetration into the membrane bilayer (Figure 12). Aromatic residues at the *N*-terminus are typical of membrane active peptides, as they are usually located at the interface between the hydrophilic and hydrophobic environments and are central for activity. A key role is likely played by the flexibility of the *N*-terminal domain, which enhances the peptide affinity for the membrane. The presence of flexibility together with a helical domain increases the peptide’s affinity to the membrane and the peptide’s stability in the membrane bilayer. The results obtained clearly show that the fungal surface potential was significantly modified by the presence of the peptide, though the permeabilization of the membrane was gradual and produced a leakage of 30%. All these data point to a local permeabilization rather than the formation of pores in the fungal membrane. In particular, the flexibility may destabilize the formation of pores, preventing the tight packing of peptides in the bundle. In contrast, flexibility facilitates the formation of self-associated helical peptides (molecular aggregates) which might be associated with a sort of carpet mechanism of interaction, where the whole membrane is covered with peptides and becomes disintegrated as it would in the presence of surfactants.

## 5. Conclusions

There is an urgent need to develop agents that effectively prevent biofilm formation and eradicate established biofilms. Herein, we presented cyclic temporin analogues that can disrupt established surface biofilms of *Candida* species at sub-MIC concentrations without being cytotoxic to mammalian cells. The data obtained show that peptide **3** bearing the lactam bridge in positions 6 and 10 emerged as the best compound. In particular, it was soluble in water at the concentrations used in the antifungal experiments, but it was aggregated in membrane bilayers mimicking fungal membranes, as also demonstrated by the change in the zeta potential of the *C. albicans* cells. Nonetheless, the analysis of all the data obtained seem to support the hypothesis of a carpet mechanism, while on the contrary, the most cytotoxic peptide (**15**), bearing the lactam bridge in the same position of peptide **3** but with a reverse orientation, also induced high leakage in the eukaryotic cells. In conclusion, the provided molecular insight into the complex behavior of the temporin analogues provides clues for the design and modification of AMPs and shows great promise as a platform for the development of novel antibiofilm agents that target biofilms and modulate the surface interactions of fungi for the treatment of chronic biofilm-associated infections.

## Figures and Tables

**Figure 1 pharmaceutics-14-00454-f001:**
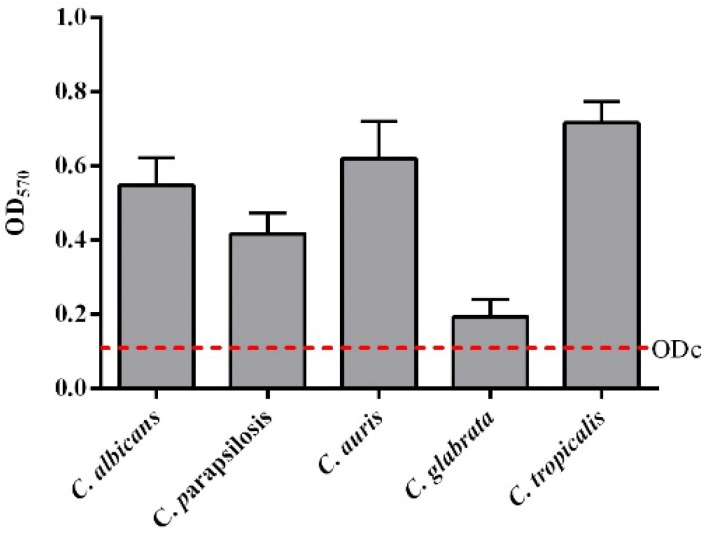
Biofilm formation capacity of five species of microorganisms using the crystal violet staining method. ODcut = mean of negative control with addition of 3 times the SD.

**Figure 2 pharmaceutics-14-00454-f002:**
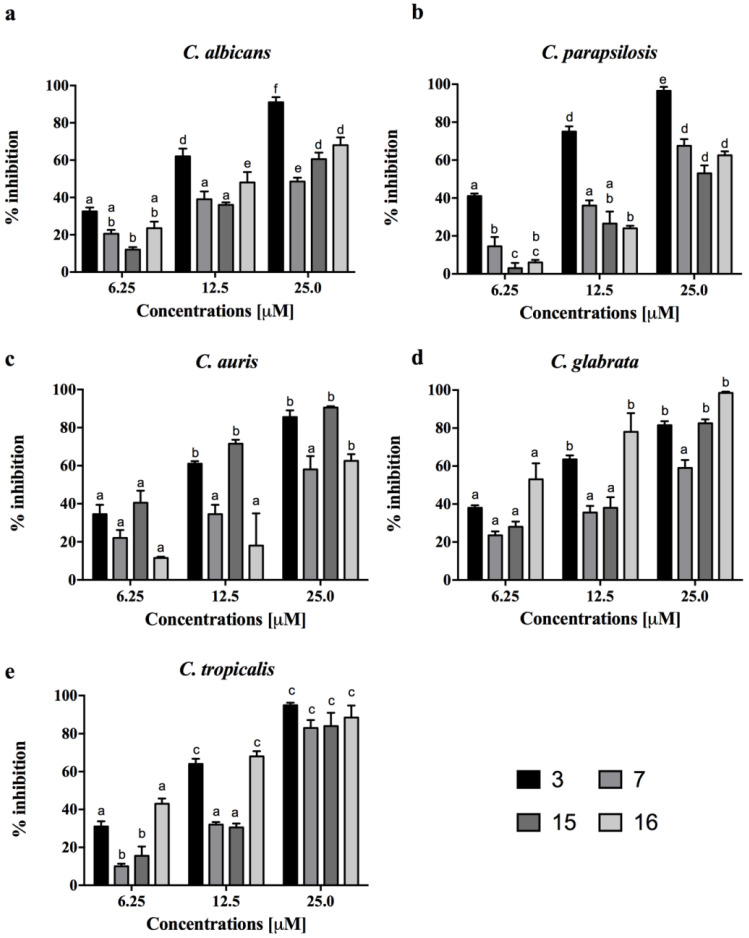
Dose-dependent (6.25, 12.5 or 25.0 μM) biofilm inhibition by peptides **3**, **7**, **15** and **16** against *C. albicans*, *C. auris*, *C. glabrata*, *C. tropicalis* and *C. parapsilosis*, quantified with crystal violet after 24 h in triplicate (**a**–**e**). Data with different letters (a–f) are significantly different (*p* < 0.05, Tukey’s).

**Figure 3 pharmaceutics-14-00454-f003:**
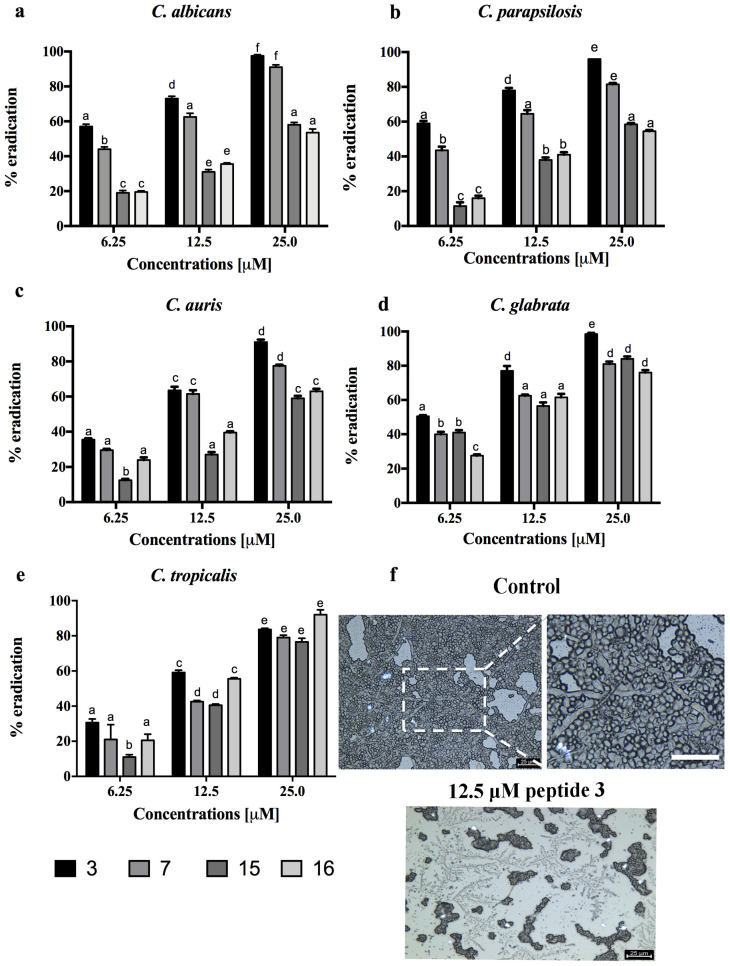
Evaluation of biomass reduction in biofilms quantified with XTT of the five *Candida* species after treatment with peptides **3**, **7**, **15** and **16** at the same sub-MIC concentrations (6.25, 12.5 or 25.0 μM) (**a**–**e**). Data with different letters (a–f) are significantly different (*p* < 0.05, Tukey’s). (**f**) The eradication effect of peptide **3** on preformed *C. albicans* ATCC 90028 biofilms was evaluated by bright field microscopy. Scale bars represent 25 μm.

**Figure 4 pharmaceutics-14-00454-f004:**
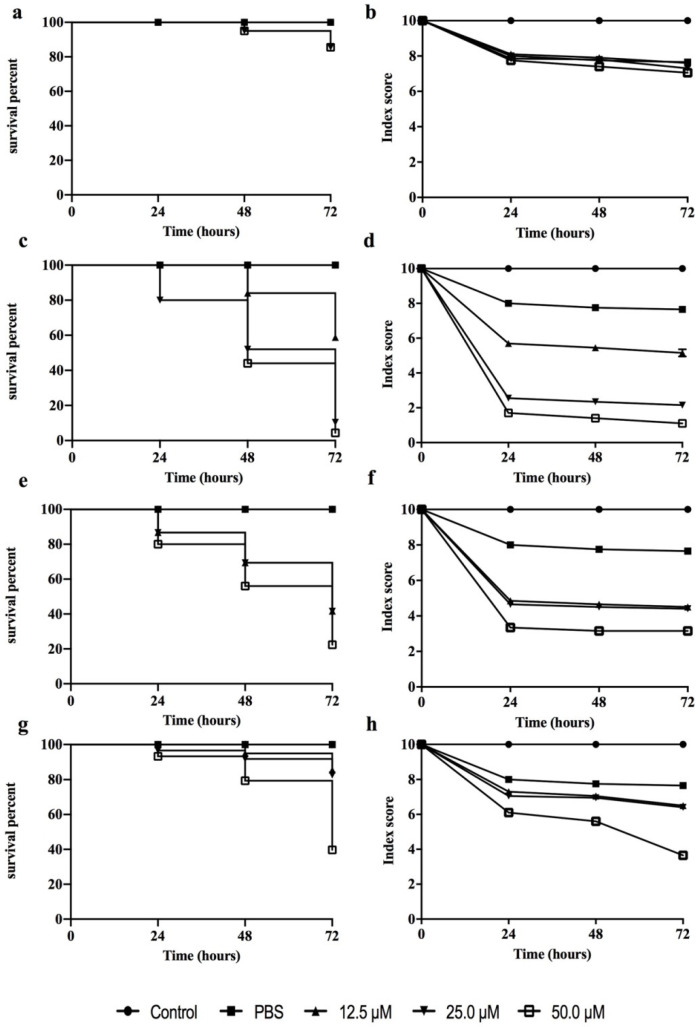
Peptide toxicity on *G. mellonella* larvae (on the left) and index score of *G. mellonella* (on the right) treated with the peptides **3** (**a**,**b**), **7** (**c**,**d**), **15** (**e**,**f**) and **16** (**g**,**h**) at the concentrations of 12.5 μM, 25.0 μM and 50 μM.

**Figure 5 pharmaceutics-14-00454-f005:**
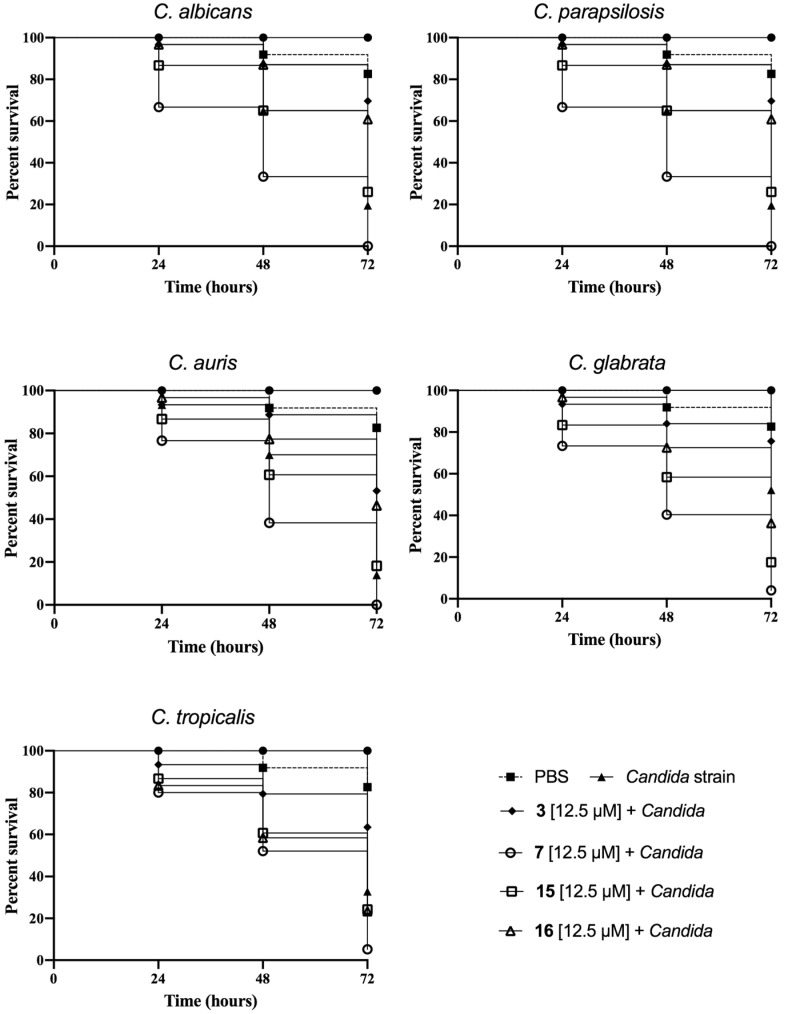
In vivo effectiveness of peptides **3**, **7**, **15** and **16** on infected *G. mellonella* larvae by different *Candida* species.

**Figure 6 pharmaceutics-14-00454-f006:**
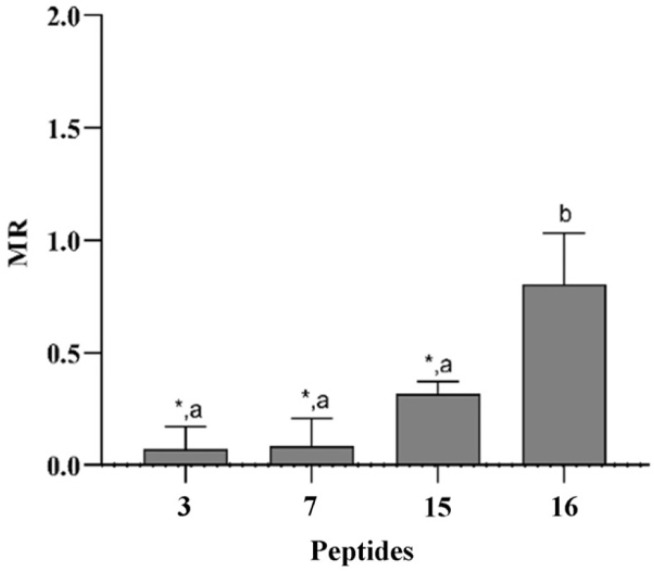
Mutagenicity ratio values of peptides **3**, **7**, **15** and **16**. The error bars represent the standard deviations. * The difference from the control group (*p* value < 0.05). Different letters show significant difference (for *p* = 0.05) according to the Tukey HSD test.

**Figure 7 pharmaceutics-14-00454-f007:**
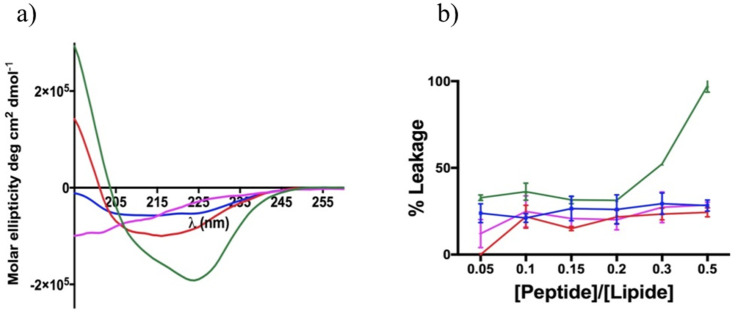
(**a**,**b**) CD spectra and membrane leakage of peptides **3**, **7**, **15** and **16** (**3** = blue line; **7** = red line; **15** = green line; **16** = pink line) in liposomes mimicking eukaryotic membranes (DOPC:Chol, 70:30, *w*/*w*).

**Figure 8 pharmaceutics-14-00454-f008:**
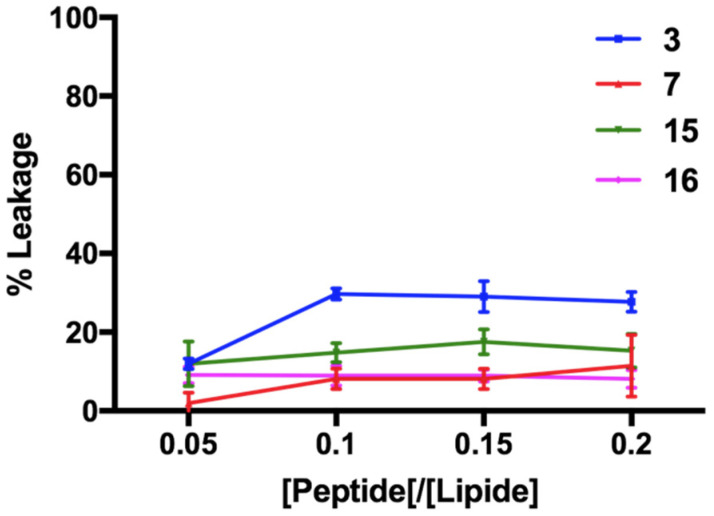
ANTS/DPX leakage by peptide in PE/PC/PI/ergosterol (5:4:1:2, *w*/*w*/*w*/*w*) LUVs.

**Figure 9 pharmaceutics-14-00454-f009:**
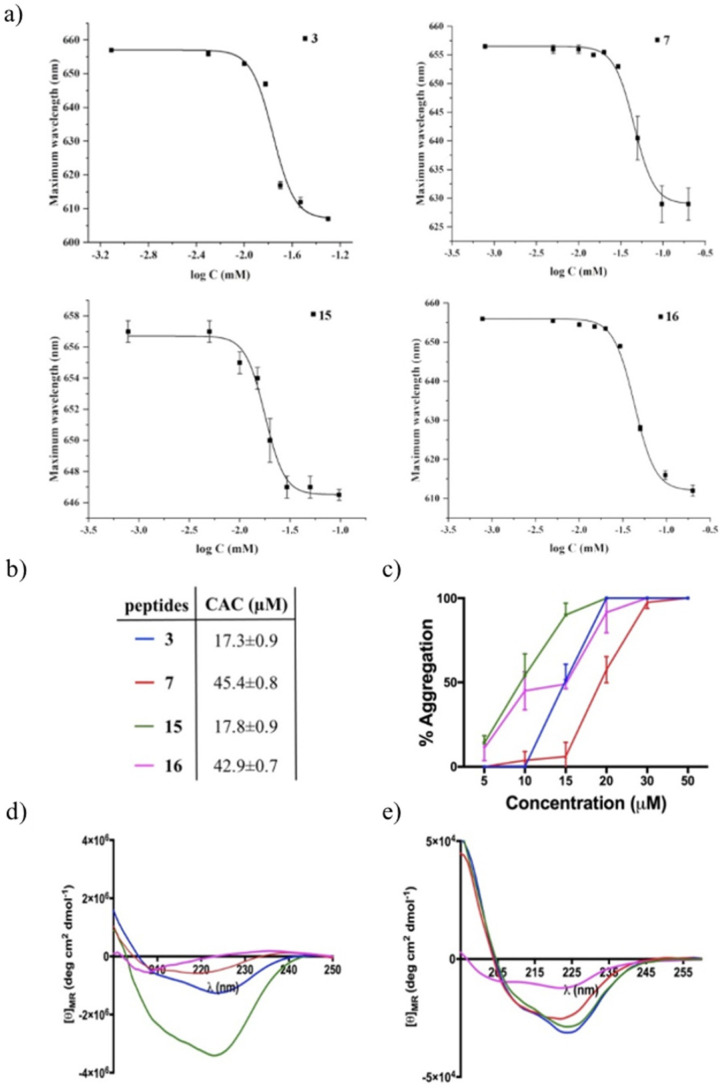
Wavelength corresponding to the maximum fluorescence emission of Nile red was plotted as a function of the concentration of peptides **3**, **7**, **15** and **16** (panel **a**), and the table (panel **b**) reports the values of the critical aggregation concentration (CAC) of peptides and the ratio of the ellipticities at 222 and 208 nm, which discriminates between monomeric (*θ*_222_/*θ*_208_ < 0.8) and oligomeric (*θ*_222_/*θ*_208_ > 1) states of the helices [48]. Peptide aggregation (**3**, **7**, **15** and **16**) monitored by ThT fluorescence (**c**) and CD spectra (**d**,**e**) in presence of liposomes mimicking fungal membranes (5:4:1:2, *w*/*w*/*w*/*w*).

**Figure 10 pharmaceutics-14-00454-f010:**
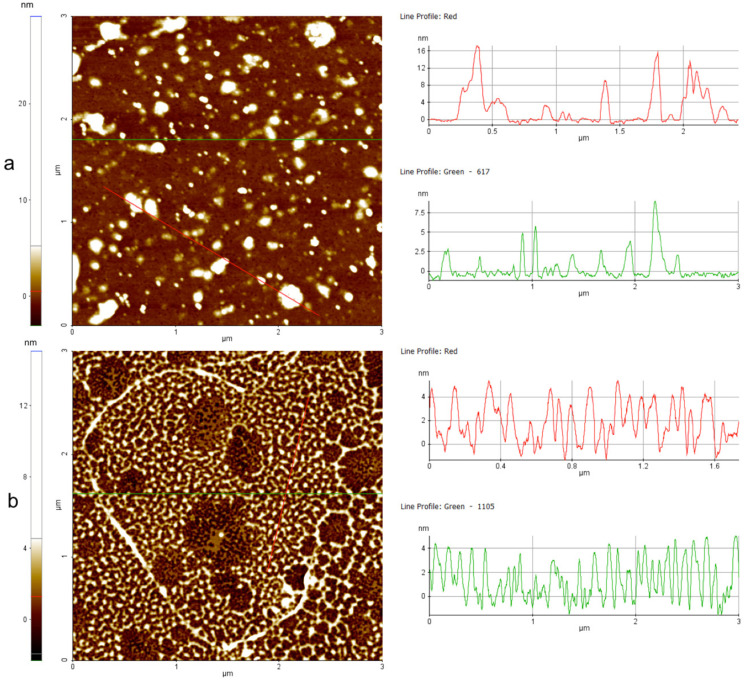
Peptide aggregation on mica substrate imaged by AFM before (**a**) and after (**b**) interaction with liposomes mimicking fungal membranes. The color scales on the left show the height measurements. In both cases, two-line profiles are reported as examples. In the absence of liposomes, peptide **3** aggregates were large (up to 300 nm wide and height up to 20 nm) (**a**). In the presence of liposomes, peptide **3** seemed to arrange on mica substrate in a kind of layer 4 nm thick, showing the strongest interaction with the substrate rather than itself, confirming a change of structure and charge in presence of liposomes.

**Figure 11 pharmaceutics-14-00454-f011:**
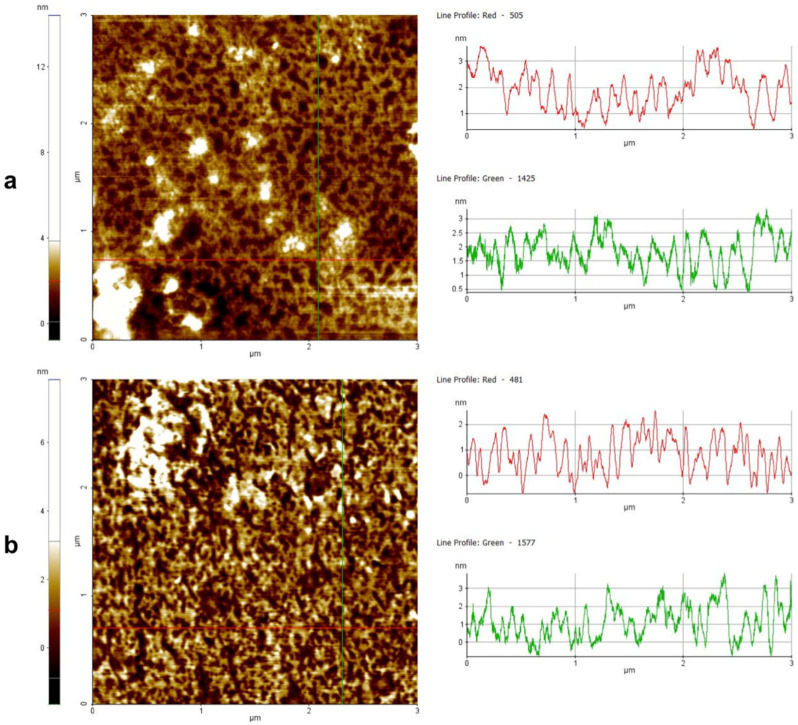
Aggregation of peptide **16** on mica substrate imaged by AFM before (**a**) and after (**b**) interaction with liposomes mimicking fungal membranes. The color scales on the left show the height measurements. In both cases, two-line profiles are reported as examples. In the absence (**a**) and presence (**b**) of liposomes, peptide **16** seemed to arrange on the mica substrate in a kind of layer 2 ÷ 3 nm thick, showing the strongest interaction with the substrate (positively charged) rather than itself (negatively charged).

**Figure 12 pharmaceutics-14-00454-f012:**
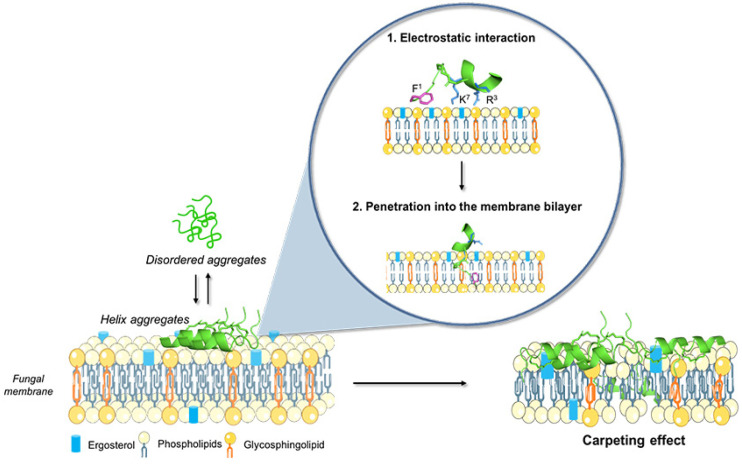
The schematic description of the hypothetical mode of action of lactam-bridged peptide **3** (PDB ID 7OSD). The peptide formed disordered aggregates in the aqueous solution while adopting an amphipathic structure in the presence of the fungal membrane. In the first step, the peptide interaction was facilitated by the electrostatic attractions between positively charged residues (Arg^3^ and Lys^7^) and anionic phospholipids featuring the membrane of yeast cells. After the peptide aggregation on the membrane, the peptide may penetrate into the membrane bilayer and cause membrane destabilization with a sort of carpet mechanism. The figure was created with BioRender.com (accessed on 20 December 2021)and smart.servier.com (accessed on 20 December 2021).

**Table 1 pharmaceutics-14-00454-t001:** Sequences of α-helical stapled peptides 1–17 [31]. The bridge involving the positions *i,i* + 4 or *i,i* + 7 is highlighted in green. Minimum inhibitory concentrations (MICs) of all peptides were determined on *C. albicans* and *C. parapsilosis* strains.

		Antimicrobial Activity (MIC, µM)	
Peptides	Sequence	*C. albicans* ATCC 90028	*C. parapsilosis* DSM 5784
**1**	F V P W F S K F **[k G R I E]**	>50	>50
**2**	F V P W F S K [**K** *l* G R **E**] L	>50	>50
**3**	F V P W F [**K** K F *l* **E**] R I L	50	50
**4**	F V P W [**K** S K F ***e***] G R I L	>50	>50
**5**	F V P [**K** F S K **E**] *l* G R I L	>50	>50
**6**	F V P W F S K F [***pra*** G R I **Az**]	>50	>50
**7**	F V P W F S K [**Pra** *l* G R **Az**] L	50	50
**8**	F V P W F [**Pra** K F *l* **Az**] R I L	>50	>50
**9**	F V P W [**Pra** S K F ***az***] G R I L	>50	>50
**10**	F V P [**Pra** F S K **Az**] *l* G R I L	>50	>50
**11**	F V P W F [**K** F *l* G R I **E****]**	>50	>50
**12**	F V P W [**K** S K F *l* G R **E]** L	>50	>50
**13**	F V P W F [**Pra** K F *l* G R I **Az**]	>50	>50
**14**	F V P W [**Pra** S K F *l* G R **Az**] L	>50	>50
**15**	F V P W F [**E** K F *l* **K**] R I L	>50	50
**16**	F V P W F [**C** K F *l* **C**] R I L	50	50
**17**	F V P W F [**S_5_** K F *l* **S_5_**] R I L	>50	>50

**Table 2 pharmaceutics-14-00454-t002:** Minimum inhibitory concentration (MIC, µM) and minimal fungicidal concentration (MFC, µM) of selected peptides against all *Candida* strains.

Peptides								
	3		7		15		16	
	MIC	MFC	MIC	MFC	MIC	MFC	MIC	MFC
*C. albicans* ATCC 90028	50	50	50	50	>50	50	50	>50
*C. parapsilosis* DSM 5784	50	50	50	50	50	50	50	>50
*C. auris* DSM 21092	50	50	50	>50	50	>50	50	>50
*C. glabrata* DSM 11226	50	50	50	50	50	>50	25	50
*C. tropicalis* DSM 11951	50	50	50	50	50	>50	25	50

**Table 3 pharmaceutics-14-00454-t003:** Zeta potential values of *C. albicans* cells after the treatment with peptides **3**, **7**, **15** and **16**.

Compound	Calculated Zeta Potential	Experimental Zeta Potential (ς, mV)
3	−3.2 ± 0.5	1.6 ± 0.3
7	−3.2 ± 0.5	−4.5 ± 1.6
15	−3.2 ± 0.5	13.5 ± 3.4
16	−3.2 ± 0.5	−6.5 ± 0.2

## Data Availability

Not applicable.

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
