# Peer review of "Antifungal and Antibiofilm Activity of Cyclic Temporin L Peptide Analogues against Albicans and Non-Albicans Candida Species"

_pharmaceutics, 2022, doi:10.3390/pharmaceutics14020454_

Round 1

Reviewer 1 Report

The manuscript is a very interesting look at the health problems of the modern world. Candida epidemiology poses a serious threat to public health, so innovative methods of searching for new solutions in the effective treatment of fungal infections are valuable. I find the study highly interesting, with an accurate selection of research methods and a comprehensible and clear presentation of the results. Correct any editorial errors in the text:

  1.   Provide the full MIC values instead of the greater-than signs (Tables 1 and 2).
  2. Correct the description under the Table 2.

Author Response

Reviewer 1

The manuscript is a very interesting look at the health problems of the modern world. Candida epidemiology poses a serious threat to public health, so innovative methods of searching for new solutions in the effective treatment of fungal infections are valuable. I find the study highly interesting, with an accurate selection of research methods and a comprehensible and clear presentation of the results. Correct any editorial errors in the text:

     Provide the full MIC values instead of the greater-than signs (Tables 1 and 2).
    Correct the description under the Table 2.

We thank the referee for his/her comments relating to Tables 1- 2, unfortunately we did not tested MIC values higher than 50 μM because not interesting for our study, we investigated only peptides with MIC under 50 μM

Reviewer 2 Report

The manuscript titled, "Antifungal and antibiofilm activity of cyclic Temporin L peptide analogs against albicans and non-albicans Candida species" by Rosa Bellavita et al., is a well articulated and detailed study on the different analogs of cyclic Temporin L. 

The manuscript is well written with interesting flow of information and findings are well explained.

Its quiet surprising to see that disulfide bridged analog 16 exhibits great antifungal activity without conformational change and its inability to bind to membranes as accessed with unchanged zeta potential of albicans cells. Did authors try to image analog 16 with AFM to look at aggregate state of the peptide or try to deduce its mechanism?

Author Response

Reviewer 2

The manuscript titled, "Antifungal and antibiofilm activity of cyclic Temporin L peptide analogs against albicans and non-albicans Candida species" by Rosa Bellavita et al., is a well articulated and detailed study on the different analogs of cyclic Temporin L.

The manuscript is well written with interesting flow of information and findings are well explained.

Its quiet surprising to see that disulfide bridged analog 16 exhibits great antifungal activity without conformational change and its inability to bind to membranes as accessed with unchanged zeta potential of albicans cells. Did authors try to image analog 16 with AFM to look at aggregate state of the peptide or try to deduce its mechanism?

We thank the reviewer for his/her suggestion and we include AFM images of peptide 16 before and after liposomes exposition in in the paper. Moreover we add the following sentences at the end of 3.11 section:

Note that, for peptide 16 AFM imaging (see figure 11) reveals no significant changes into the arrangement states on mica substrate before and after interaction with liposomes. This result confirms that no significant alteration in the charge of peptides 16 (see zeta potential results) was obtained before and after liposomes exposition. In fact, in both cases the strong interaction between the negatively charged peptides 16 and the positively charged substrate results in an arrangement on mica substrate in form of thin layer (2 ÷ 3 nm).

Reviewer 3 Report

The work is devoted to the actual topic of studying antimicrobial peptides in the aspect of its structural and functional. These substances considered as fungicides and have potential as therapeutic agents.

The main remarks:

  1. Why do the authors use the XTT assay to establish the eradication capacity of the studied peptides? (figure 3). Usually XTT assay is used to evaluate metabolic activity of cells, but not biofilm biomass (as follows from the legend in Figure 3). For this purpose, it is more correct to use the CV assay, since the action of peptides just decrease in cells metabolic activity, but this is no relevant to the eradication aspect of its activity. So, authors should assess biofilm eradication using a CV assay or rewrite the corresponding parts of the manuscript in appropriated manner.
  2. Moreover, in the figure 3f, hyphae are not visible in the control sample, so it is not clear what exactly the authors mean in the sentences (lines 460-463, and 719). Actually, I think the authors don’t have been able to form true candida biofilms. This can be seen from the micrograph (Figure 3f), where a monolayer of adherent cells is visualized. The authors can see what a Candida biofilm looks like in the works, for example (doi/10.2217/nnm-2018-0183 ; 10.3389/fmicb.2018.02892).
  3. According to the measurement of the zeta potential. I would like to see the following data: the charge (calculated) of the peptides themselves, the order of change in the zeta potential of fungal cells after the treatment with peptides (instead of final Z-values). You could to modify the Table 3 as follows (as an example): Column 1 is peptide (charge) + cells (Z-value of intact cells, mV) > column 2 is treated cells (z-value of treated cells, mV).
  4. Lines 695-719. Need to rewrite the Discussion part as it contains a lot of descriptive information that is more appropriate in the Results section.It would be better to talk about the structure of the peptides (why there are lactam bridge and 1,4-triazolic bridge) and how the peptide structure affects the revealed activity, including fungicidal and anti-biofilm (sub-MIC effects).
  5. Section5. It is better to move the description of XTT assay to the section where it is used, since I did not find data on the metabolic activity of the formed biofilms (in the part of the results that examines the ability of strains to biofilm formation).

Minor remarks

Table 2. Check the MIC value for peptide 15. MIC do not be a higher than MFC

Line 329. Please, define AFM abbreviation since it is the first introduction

Line 151.  106 cells/mL or 10E6 cells/mL ?

Lines 160, 173.  cells/mL instead cell/mL

Lines 167-168. MFC definition repeated twice.

Lines 441-442. Clarify that is "ODc"

Line 447. What exactly means “Free planktonic microorganisms”

Author Response

Reviewer 3

The work is devoted to the actual topic of studying antimicrobial peptides in the aspect of its structural and functional. These substances considered as fungicides and have potential as therapeutic agents.

The main remarks:

1. Why do the authors use the XTT assay to establish the eradication capacity of the studied peptides? (figure 3). Usually XTT assay is used to evaluate metabolic activity of cells, but not biofilm biomass (as follows from the legend in Figure 3). For this purpose, it is more correct to use the CV assay, since the action of peptides just decrease in cells metabolic activity, but this is no relevant to the eradication aspect of its activity. So, authors should assess biofilm eradication using a CV assay or rewrite the corresponding parts of the manuscript in appropriated manner.

We thank the reviewer for his comments. We now better explained in the text (see paragraph 2.5) that the evaluation of the total biofilm biomass was performed using the CV staining method. Furthermore we used the XTT test for the determination of the biomass after eradication. Indeed, in literature is reported the use of the XTT assay to evaluate the vital biomass after eradication and we also did this in our previous studies (Microbial Pathogenesis 125 (2018) 189; Pathogens 10 (2021) 214; Applied end Environmental Microbiology. 87 (2021) :e00391-21)

2. Moreover, in the figure 3f, hyphae are not visible in the control sample, so it is not clear what exactly the authors mean in the sentences (lines 460-463, and 719). Actually, I think the authors don’t have been able to form true candida biofilms. This can be seen from the micrograph (Figure 3f), where a monolayer of adherent cells is visualized. The authors can see what a Candida biofilm looks like in the works, for example (doi/10.2217/nnm-2018-0183 ; 10.3389/fmicb.2018.02892).

We thank the referee for his comment. Indeed we realized that the control image was not representative because of the fact that with our magnification the hyphae were not clearly visible. For that reason we add an image with a zoom on a group of hyphae

3. According to the measurement of the zeta potential. I would like to see the following data: the charge (calculated) of the peptides themselves, the order of change in the zeta potential of fungal cells after the treatment with peptides (instead of final Z-values). You could to modify the Table 3 as follows (as an example): Column 1 is peptide (charge) + cells (Z-value of intact cells, mV) > column 2 is treated cells (z-value of treated cells, mV).

We thank referee for his/her suggestion and modified accordingly. Now the Table evidences more clearly the effect of the peptides on the zeta potential.

4. Lines 695-719. Need to rewrite the Discussion part as it contains a lot of descriptive information that is more appropriate in the Results section. It would be better to talk about the structure of the peptides (why there are lactam bridge and 1,4-triazolic bridge) and how the peptide structure affects the revealed activity, including fungicidal and anti-biofilm (sub-MIC effects).

We revised the Discussion section considering Reviewer’s comments. In particular, we explored specific positions of the original peptide sequence and introduced bridges of different chemistry (e.g., lactam- and 1,4-triazolic-bridges) based on their propensity to stabilize the a-helical conformation (please, see 31). Also, we believe the aggregation is the key phenomenon that justify the fungicidal activity and related mechanism of action. Other investigations useful to correlate the peptide structure with fungicidal and anti-biofilm activities (e.g., NMR acquired in membrane environments) were behind the scope of this work. Anyway, we thank the reviewer for his/her comments that will be taken in consideration for our future studies.

5. Section5. It is better to move the description of XTT assay to the section where it is used, since I did not find data on the metabolic activity of the formed biofilms (in the part of the results that examines the ability of strains to biofilm formation).

We agree with referee suggestion, we moved the XTT assay to the right section, see lines 194-199

Round 2

Reviewer 3 Report

The authors tried to take into account my comments and made the appropriate corrections.